# Quercetin Impairs the Growth of Uveal Melanoma Cells by Interfering with Glucose Uptake and Metabolism

**DOI:** 10.3390/ijms25084292

**Published:** 2024-04-12

**Authors:** Aysegül Tura, Viktoria Herfs, Tjorge Maaßen, Huaxin Zuo, Siranush Vardanyan, Michelle Prasuhn, Mahdy Ranjbar, Vinodh Kakkassery, Salvatore Grisanti

**Affiliations:** Department of Ophthalmology, University of Lübeck, Ratzeburger Allee 160, 23562 Luebeck, Germany; v.herfs@gmx.de (V.H.); tjorge.maassen@web.de (T.M.); zuohuaxin@163.com (H.Z.); sirivardanyan@gmail.com (S.V.); michelle.prasuhn@uksh.de (M.P.); vinodh.kakkassery@uni-luebeck.de (V.K.); salvatore.grisanti@uksh.de (S.G.)

**Keywords:** uveal melanoma, glycolysis, pentose phosphate pathway, quercetin, glucose uptake, proliferation, oxidative stress

## Abstract

Monosomy 3 in uveal melanoma (UM) increases the risk of lethal metastases, mainly in the liver, which serves as the major site for the storage of excessive glucose and the metabolization of the dietary flavonoid quercetin. Although primary UMs with monosomy 3 exhibit a higher potential for basal glucose uptake, it remains unknown as to whether glycolytic capacity is altered in such tumors. Herein, we initially analyzed the expression of *n* = 151 genes involved in glycolysis and its interconnected branch, the “pentose phosphate pathway (PPP)”, in the UM cohort of The Cancer Genome Atlas Study and validated the differentially expressed genes in two independent cohorts. We also evaluated the effects of quercetin on the growth, survival, and glucose metabolism of the UM cell line 92.1. The rate-limiting glycolytic enzyme *PFKP* was overexpressed whereas the *ZBTB20* gene (locus: 3q13.31) was downregulated in the patients with metastases in all cohorts. Quercetin was able to impair proliferation, viability, glucose uptake, glycolysis, ATP synthesis, and PPP rate-limiting enzyme activity while increasing oxidative stress. UMs with monosomy 3 display a stronger potential to utilize glucose for the generation of energy and biomass. Quercetin can prevent the growth of UM cells by interfering with glucose metabolism.

## 1. Introduction

Despite the advances in the local control of primary uveal melanoma (UM), approximately half of UM patients develop metastases, mainly in the liver, which sadly results in a very short average survival time of less than six months [1,2,3]. The risk of metastasis is particularly high for primary UM cells that exhibit the loss of one copy of chromosome 3 (monosomy 3) [4,5]. Similarly, the presence of monosomy 3 in the metastasized UMs has been associated with more rapid disease progression [6]. However, the molecular mechanisms and pathophysiological factors that give rise to aggressive UM metastases remain incompletely understood, which impedes the development of efficient therapies.

Interestingly, UM patients exhibited an insulin-resistant serum profile with a slight but significant increase in their fasting glucose levels compared to age-matched controls [7]. Moreover, the basal level of glucose uptake in UM cells may be influenced by chromosome 3 status, as suggested by the increased metabolic activity of monosomy 3 tumors in fluorodeoxyglucose positron emission tomography scans [8,9]. We have also recently reported the upregulation of the high-affinity glucose transporter GLUT1 in primary UMs with monosomy 3, possibly as a compensation for the very-low-affinity glucose transporter GLUT2 [10], which is encoded by the *SLC2A2* gene on chromosome 3 [11]. These findings therefore highlight hyperglycemia and the increased potency for glucose influx as novel pathophysiological factors that may aggravate the course of UM. Yet, it is still not well known as to how glucose is mainly processed inside UM cells and whether the presence of monosomy 3 confers a growth advantage through the modulation of glucose metabolism to address cellular demands more efficiently.

Proliferating cells need to have high biosynthetic activity to increase their biomass and replicate their genome. The cells should therefore maintain the influx of oxygen and nutrients at a sufficient rate to generate energy in the form of adenosine triphosphate (ATP) and conduct biosynthetic reactions. However, the rapidly proliferating solid malignancies may overgrow their blood supply, with them becoming progressively deprived of oxygen and nutrients. Cancer cells adapt to such conditions by preferentially executing glycolysis instead of oxidative phosphorylation even in the presence of oxygen, which is known as the Warburg effect [12,13,14,15,16,17,18].

The production of ATP via glycolysis may initially appear to be an inefficient form of metabolism due to the lower energetic yield obtained from each glucose molecule compared to oxidative phosphorylation [15,17,18]. However, glycolysis can be completed at a rate that may be up to 100 times faster than oxidative phosphorylation [19]. Glycolysis can therefore enable the rapid production of sufficient ATP as long as the cells are provided abundantly with glucose [17]. In addition, glycolysis generates metabolic intermediates that can be diverted into the “pentose-phosphate-pathway” (PPP) in a reversible manner for the production of biosynthetic materials depending on cellular needs [17].

The PPP is divided into branches that produce the nucleotide precursor ribose-5-phosphate and the cofactor NADPH, which serves as a reducing agent during the biosynthesis of lipids and amino acids. NADPH also enables the regeneration of glutathione, which is essential for protection against the reactive oxygen species that are mainly produced during oxidative phosphorylation [17,19,20,21]. Tumor cells may exploit these pathways as suggested by the overexpression of the rate-limiting enzymes of glycolysis and the PPP in malignant diseases such as lung, liver, colorectal, prostate, breast, and cervical cancer [17,20,21]. Despite the association between several glycolysis-associated genes and poor survival in UM [22,23,24], it is not known whether the potential for the glycolytic switch is altered in monosomy 3 tumors and whether the glycolysis–PPP axis in UM cells can be modulated by natural compounds.

Flavonoids are bioactive plant molecules that are present in a wide variety of foods and drinks such as fruits, vegetables, herbs, spices, teas, and wine. Quercetin is the most abundant dietary flavonoid, with many of its natural sources being included in the Mediterranean diet [25,26]. Quercetin is able to exert beneficial effects against inflammation, insulin resistance, and hyperglycemia in animal models [27,28]. Quercetin also demonstrated promising anti-carcinogenic potential by suppressing the metastatic activities of various tumor cells [25,26,29,30,31] while protecting normal cells from the side effects of chemotherapy and radiotherapy in preclinical studies [29,31]. The cytotoxic effect of quercetin was higher in the aggressive tumor cells compared to the slowly growing ones [30]. In addition, quercetin was able to inhibit aerobic glycolysis in diverse tumor cells [18,32,33] and suppress the self-renewal capacity of cancer stem cells [31]. Despite its well-established antioxidant effect, quercetin can also function as a pro-oxidant at high concentrations by depleting the glutathione reserves in cancer cells [18,34]. However, it remains unknown whether quercetin can suppress the growth of UM cells.

In this study, we initially determined the expression profile of the major genes involved in glycolysis and the PPP in the UM cohort of The Cancer Genome Atlas (TCGA) Study and validated the differentially expressed genes in two independent, publicly available cohorts. We also analyzed the growth, survival, and glucose metabolism of the UM cell line 92.1 in response to quercetin.

## 2. Results

### 2.1. Expression of the Genes Involved in Glycolysis and the PPP with Regard to Monosomy 3 Status and Metastases in the UM-Cohort of the TCGA Study

We initially assembled a list of *n* = 151 human genes that are involved in glycolysis and the PPP using the Gene Ontology (GO) and Reactome databases. The gene list also included the “lactate (transmembrane) transport” (Appendix A).

In the UM cohort of the TCGA study, a total of *n* = 67 genes (44.4%) were differentially expressed with regard to the copy number of chromosome 3, with *n* = 27 genes (17.9%) retaining significance after multiple hypothesis testing (Appendix A). Among the latter group, *n* = 5 genes were also significantly associated with the development of metastases (*p*-adjusted < 0.05, Figure 1A, Appendix A). The genes that were upregulated in the monosomy 3/metastatic tumors (*PFKP*, *NUP88*) exhibited an average fold change (FC) of 2.06 ± 0.82, whereas the downregulated genes (*INSR*, *RBKS*, and *ZBTB20*) differed by an average of −1.87 ± 0.44-fold (Figure 1B).

An unbiased gene set enrichment was performed for the differentially expressed genes. The most relevant biological processes, pathways, and phenotypes involved the “positive regulation of the glycolytic and purine nucleotide metabolic pathways”, “phosphofructokinase activity”, “insulin-like growth factor II/I/AMP binding”, “Pentose phosphate/HIF-1/AMPK pathways”, “galactose, fructose, and mannose metabolism”, “insulin receptor substrate activation”, “signal attenuation/insulin receptor recycling”, and “diabetes (type 2)” (Figure 1C).

### 2.2. Validation of the Differentially Expressed Genes

Validation of differential gene expression was performed using the normalized microarray data of two independent studies which are publicly available in the Gene Expression Omnibus (GEO) database. The first study involved the total RNA isolated from the primary tumors of *n* = 63 UM patients (*n* = 24 females; *n* = 39 males) with an average age of 61.0 ± 12.3 years (GEO accession number: GSE22138). Systemic metastases developed in *n* = 35 of these patients (55.6%). The second microarray study included the primary tumors of *n* = 57 patients with *n* = 25 females and *n* = 32 males. Metastases were detected in 56.1% (*n* = 32) of these patients (GEO accession number: GSE44295).

Expression of the *PFKP* and *ZBTB20* genes was significantly altered with regard to the metastases in the validation cohorts. *PFKP* (locus: 10p15.2) was positively associated with the development of metastases, whereas the *ZBTB20* gene (locus: 3q13.31) was downregulated in the metastatic tumors (*p*-raw < 0.05, Figure 2A).

Further analysis of the validated genes revealed significant associations between the expression of *PFKP* or *ZBTB20* and the survival rate and multiple prognostic factors in the UM cohort of the TCGA study. The upregulation of *PFKP* and the downregulation of *ZBTB20* were correlated with shorter overall and disease-specific survival, an epithelioid morphology, heavy pigmentation, and the presence of closed loops. *PFKP* was also positively associated with tumor diameter and extrascleral extension whereas the tumors with *ZBTB20* deficiency exhibited a more diffuse rather than focal pattern of macrophage infiltration (Figure 2B,C).

### 2.3. Anti-Proliferative and Pro-Oxidant Effects of Quercetin on the Cultured UM Cells

Based on the findings of our gene expression analysis, which highlighted glucose metabolism as a crucial factor that may be involved in the aggressive growth of UM cells, we next focused on the inhibitory potential of quercetin on metabolic activity, proliferation, survival, and redox state in the UM cell line 92.1. Incubation of the 92.1 cells with quercetin for three days induced a significant, dose-dependent reduction in the number of metabolically active cells, with an IC50 of quercetin at 44.05 µM (Figure 3A,B, *p* < 0.001 for all groups, Kruskal–Wallis test). This effect was associated with the down-regulation of the proliferation marker Ki67 and a 69% decline in the incorporation of BrdU in response to 50 µM quercetin compared to the solvent control (Figure 3C,D, *p* < 0.05 for all groups, Kruskal–Wallis test). Incubation with 50 µM quercetin also reduced the ratio of live/dead cells by approximately 36% (Figure 3E,F, *p* < 0.01 for all groups, one-way analysis of variance) and increased the accumulation of reactive oxygen species (ROS) by two-fold (Figure 3G,H, *p* = 0.049, Mann–Whitney U test) compared to the solvent control.

### 2.4. Suppression of Glucose Uptake and Metabolism in the UM Cells in Response to Quercetin

Having determined the optimal concentration of quercetin for the suppression of metabolic activity in the 92.1 cells, we next evaluated the outcomes of quercetin treatment on different aspects of glucose metabolism, such as glucose intake, glycolytic rate, ATP production, PPP activity, and glutathione levels. Incubation of the 92.1 cells with 50 µM quercetin made it possible to significantly reduce the uptake of the fluorescent glucose analog 6-NBDG by approximately 32–36% compared to the solvent or untreated controls (*p* = 0.041, Kruskal–Wallis test, Figure 4A,B). Quercetin could also suppress the glycolytic rate by approximately 45–48% within 3 h and ATP production by 21–25% after 2 days compared to the untreated and solvent controls (*p* = 0.010 and *p* = 0.009, respectively, one-way analysis of variance, Figure 4C,D). The activity of the glucose-6-phosphate dehydrogenase (G6PDH), which serves as the rate-limiting enzyme of the PPP [17,20,21], underwent an approximately 20–22% decrease in response to 50 µM quercetin compared to the untreated and solvent controls after 11–13 h (*p* < 0.05, Mann–Whitney U test with respect to the untreated or solvent controls, Figure 4E). We also observed a slight but significant decline of 8.4% in the levels of total glutathione after incubation with 50 µM quercetin for 7–10 h compared to the untreated or solvent controls (*p* < 0.05, Mann–Whitney U test, Figure 4F).

### 2.5. Outcomes of Quercetin Treatment on the Expression of the PFKP and ZBTB20 Proteins in the UM Cells

To elucidate whether the validated, glucose-related genes are involved in the suppressive effects of quercetin on glucose metabolism, we also analyzed the expression of the PFKP and ZBTB20 proteins via immunocytochemistry and blotting in the UM cells that were incubated with or without 50 µM quercetin or DMSO for 1 day. Our results demonstrated uniform expression of both proteins regardless of the treatment, suggesting that quercetin does not exert a significant influence on the levels of these proteins (Figure 5).

## 3. Discussion

The prevention of lethal UM metastases remains an unresolved clinical need, which sadly leaves affected patients with a very short survival time of several months. In this study, we provide the first evidence that primary UMs with higher metastatic potential exhibit a greater glycolytic gene expression profile, whereas the incubation of the UM cell line 92.1 with the dietary flavonoid quercetin could suppress cell growth by interfering with glucose uptake and metabolism.

Our gene expression analysis demonstrated the overexpression of *PFKP* in the aggressive UM samples from three independent cohorts, whereas the *ZBTB20* gene was significantly downregulated in the tumors that developed metastases. The *PFKP* gene on chromosome 10p15.2 encodes the platelet isoform of phosphofructokinase, which serves as the most important rate-limiting enzyme of glycolysis by catalyzing the irreversible phosphorylation of fructose-6-phosphate to fructose-1,6-biphosphate [35,36]. Moreover, the silencing of *PFKP* resulted in a reduction in PPP activity and nucleotide biosynthesis in renal cancer cells [36]. The overexpression of *PFKP* may therefore be providing the UMs with monosomy 3 a growth advantage by channeling the glucose metabolites towards glycolysis or the PPP depending on the metabolic demands and cellular priorities. The phosphofructokinase isoforms are also upregulated in diverse tumors and regarded as potential therapeutic targets [35,36,37]. Enhanced glycolytic metabolism was also detected in the T cells with deficiency in the transcriptional repressor ZBTB20 [38], which is encoded by a gene on chromosome 3q13.31. However, there is conflicting evidence regarding the function of ZBTB20 in tumorigenesis, with the overexpression of this protein noted in hepatocellular carcinoma [39]. ZBTB20 could also promote the invasion of various tumor cells, including glioblastoma and breast and gastric cancers [40,41,42]. Further research is therefore required to elucidate the functional consequences of *ZBTB20* downregulation in UMs with monosomy 3. Interestingly, the silencing of *Zbtb20* was associated with a slight increase in the expression of phosphofructokinase in mouse liver cells, which failed to reach significance [43]. It would therefore be interesting to analyze whether the deficiency in *ZBTB20* mRNA promotes the overexpression of *PFKP* in UM cells or whether the upregulation of *PFKP* is a response to the higher rate of glucose influx into monosomy 3 tumors.

Our preliminary findings also demonstrate the anti-proliferative and toxic effects of the dietary flavonoid quercetin in the UM cell line 92.1 via interference with glucose metabolism. For instance, the extent of glucose uptake was lowered by approximately 25–30%, whereas the rate of glycolysis underwent an almost 50% reduction in response to quercetin. The more profound inhibitory effect of quercetin on the latter event suggests that this flavonoid may be interfering with certain glycolytic enzymes or their regulators. However, we did not observe a significant change in the expression of the PFKP and ZBTB20 proteins in response to quercetin. It therefore remains to be determined whether the activity rather than the total levels of these proteins or other glycolytic enzymes is influenced by quercetin in UM cells. The suppressive effects of quercetin on glucose metabolism also deserve further analysis with regard to the paralogous genes *GNAQ* or *GNA11*, which undergo activating mutations in 80–90% of UMs [44,45]. The somatic mutations in *GNAQ*/*GNA11*, which usually occur in a mutually exclusive pattern, can initiate UM tumorigenesis via the constitutive activation of the alpha-subunits of the heterotrimeric G proteins Gq or G11, respectively [44,45]. Remarkably, oncogenic Gq/G11 signaling was recently reported as the major driver of metabolic reprogramming in UM cells via the promotion of glucose uptake, glycolysis, and mitochondrial respiration [45]. Since 92.1 cells harbor the Q209L-activating mutation in *GNAQ* [46], it would be very interesting to evaluate the inhibitory potential of quercetin on the aberrant Gq signaling in UM in future studies.

Our study entails several limitations, such as the administration of quercetin at the single dose of 50 µM in the majority of our experiments. Although this dosage was selected as the optimal concentration based on the screening of metabolic activity via the MTT assay, it may not have been sufficient to interfere with all of the cellular events that we analyzed at the same efficacy. Moreover, the incubation period with quercetin varied from 30 min to three days among the different assays, which complicated the interpretation of the treatment outcomes. Three-day incubation was preferred to evaluate the long-term effects of quercetin on metabolic activity, proliferation, and survival. For the remaining assays, we incubated the cells for shorter periods based on the assay principle and/or previous data regarding the influence of quercetin on the cellular event of interest. For instance, the glucose uptake and glycolytic rate were evaluated through exposure of the cells to the test conditions for 30 min or 3 h, respectively, based on the recommendations of the assay manufacturers. To analyze G6PDH activity, glutathione reserves, and ROS accumulation, we preferred the incubation periods of 7–13 h based on a previous study, which reported the time-dependent effects of quercetin on glutathione levels in human aortic endothelial cells. In the aforementioned work, quercetin resulted in a temporal increase in the ratio of total glutathione (GSH) to the oxidized form (GSSG), which reached a peak after 6 h despite the significant decrease in the levels of both the total and oxidized glutathione [47]. Regarding the ATP reserves, we selected an incubation time of 2 days to gain insight into the energetic status of UM cells after long-term exposure to quercetin. For this assay, we also needed to normalize the results based on the amount of cellular material, which deviated substantially among the treatment groups owing to differences in the proliferation and survival rate over extended periods. However, we could not evaluate the protein concentration of the cellular lysates as a control for equal cellular material due to the interference of the lysis buffer of the ATP assay with the protein quantification reagents, as stated by the manufacturer. We therefore quantified the number of cells in each group for the normalization of the ATP levels. Based on these factors, we considered the incubation period of 2 days more appropriate to reduce the confounding effects of proliferation and survival in the ATP assay while providing the cells sufficient time to respond to the test conditions. A further limitation of our study is the utilization of a single cell line. Future studies with validated UM cell lines from different donors as well as normal control cells such as choroidal melanocytes, fibroblasts, or normal skin cells would therefore provide deeper insight into the therapeutic potential and safety of quercetin for the management of UM. The time- and dose-dependent effects of quercetin treatment on different aspects of glucose metabolism also deserve further investigation.

Quercetin is usually available as glycosylated isoforms which are conjugated to sugars like glucose or rutinose. Earlier studies have reported that quercetin glycosides can compete with glucose to pass through the energy-dependent or -independent glucose transporters SGLT-1 or GLUT1, respectively, interfering with glucose influx [48,49]. The free, aglycone form of quercetin, which we used in our study, could also reduce glucose uptake in Caco-2 cells, although to a lesser extent compared to the glycosylated isoforms [50]. It would therefore be very interesting to determine the glucose transporters that are targeted by the quercetin aglycone in UM cells in future studies, with particular focus on the GLUT family of energy-independent transporters [11,51].

Our findings further demonstrate a reduction in ATP production of approximately 22% in response to quercetin. The magnitude of this effect also remained lower than the 50% decrease in the glycolytic rate, suggesting that the quercetin-treated UM cells may be switching to mitochondrial metabolism to compensate for the energy shortage. However, an earlier study reported that UM cells with monosomy 3 exhibit a higher capacity for mitochondrial function [52]. A certain level of mitochondrial respiration could also promote the growth of UM cells [53]. Although these findings may initially appear to contradict our results, hyperactivity of mitochondria for prolonged periods can intensify oxidative stress [54]. This may also account for the almost two-fold increase in the levels of reactive oxygen species, which mainly arise as by-products of oxidative phosphorylation, in the quercetin-treated UM cells in our study. Although low concentrations of reactive oxygen species may promote cell proliferation by acting as second messengers, the massive accumulation of oxidative stress may overwhelm the repair capacity of cells, resulting in growth arrest or cell death [55]. Interestingly, the reactive oxygen species and mitochondrial activity were diminished in mesothelioma cells or human fibroblasts, respectively, with the inactivating mutations of the tumor suppressor *BAP1* [56,57]. Deficiency in BAP1, which is encoded by a gene on chromosome 3, is a significant prognostic factor which is associated with a higher metastatic risk in UM patients [5]. The influence of BAP1 depletion and quercetin treatment on the mitochondrial activity of UM cells therefore merits future investigation.

In our study, the activity of G6PDH, which serves as the rate-limiting enzyme of the PPP [17,20,21,58], underwent an approximately 30% decline in response to quercetin. Interference with PPP activity may also account for the mild but significant decrease of approximately 8.4% in the levels of total glutathione after quercetin treatment. The PPP plays a fundamental role in the cellular defense against oxidative stress by enabling the production of the co-factor NADPH, which acts as a reducing agent for the regeneration of glutathione [58]. However, NADPH can also be produced by alternative mechanisms that are mainly catalyzed by mitochondrial enzymes [59]. This compensation may therefore account for the modest inhibitory effect of quercetin on the levels of total glutathione as opposed to PPP activity, which deserves further examination.

In our present work, the effective dosage of quercetin on the 92.1 cell line was relatively high, with an IC50 of 44.1 µM. This concentration was in accordance with the findings of earlier studies, which reported the anticarcinogenic effects of quercetin at the optimal doses of approximately 20–160 µM in diverse tumor cell lines derived from ovarian, breast, and hepatocellular carcinomas [60,61,62]. In our study, quercetin was administered in RPMI-1640 medium, which is hyperglycemic with a glucose concentration of 11 mM. To simulate an insulin-resistant rather than a diabetic environment, UM cells can be maintained under milder or transiently hyperglycemic conditions. To mimic the regular intake of a quercetin-rich diet in vitro, UM cells can also be exposed to lower doses of quercetin for longer periods in future studies.

The bioavailability of quercetin has remained a concern due to the low solubility and absorption of this flavonoid. After ingestion, the lipophilic aglycone form of quercetin can diffuse passively from the intestinal lumen through the enterocytes, with it being absorbed into the hepatic portal vein. In contrast, the lipophobic glycosides of quercetin need to be converted to the aglycone form in the intestinal lumen or enterocytes before being absorbed into the hepatic portal vein. The quercetin aglycone then undergoes further metabolization in the liver before being distributed to other body tissues [63]. Interestingly, the hepatic portal vein may also serve as the major entry route of the disseminated UM cells into the liver, because the liver receives the majority of its blood supply via this vessel [64]. The regular intake of a quercetin-rich diet would therefore increase the abundance of this flavonoid in the portal vein and liver, which may create a less permissive microenvironment for the hepatic micrometastasis of UM.

To enhance the bioavailability of quercetin, this flavonoid can also be administered as a dietary supplement, which usually range between doses of 150 and 5000 mg [63]. The intake of 150–730 mg quercetin aglycone per day could indeed significantly reduce the blood pressure of hypertensive individuals [65]. A supplement of up to 5000 mg of quercetin per day for 4 weeks was also reported to be safe without adverse effects in clinical studies [63,66]. Assuming the complete absorption and metabolization of the ingested quercetin (molecular weight: 302.236 g/mole), the daily intake of 76 mg of quercetin would be necessary to reach a plasma concentration of 50 µM in a total blood volume of 5 L. It may therefore be feasible to reach the optimal concentration of quercetin through dietary supplements, which needs to be verified in future studies with animal models of UM metastasis.

## 4. Materials and Methods

### 4.1. Analysis of Gene Expression

The list of genes involved in glycolysis and/or the PPP was constructed using the Gene Ontology (http://geneontology.org/) and Reactome (https://reactome.org/) databases (accessed on 21 April 2021). The mRNA expression data of the UM cohort of the TCGA study were downloaded from the Xena platform of the University of California, Santa Cruz (https://xena.ucsc.edu/; accessed on 23 April 2021). Validation of gene expression was performed using the normalized microarray data of two independent studies available in the GEO database (https://www.ncbi.nlm.nih.gov/geo/; accession numbers: GSE22138 and GSE44295; accessed on 10 May 2021). Gene expression was presented in log_2_-converted values. Fold changes were calculated from the mean expression values. Heatmaps were generated using the z-scores of gene expression. Kaplan–Meier curves were constructed using the UCSC Xena platform. The unbiased analysis of gene set enrichment was performed by using the Enrichr database [67].

### 4.2. Cell Culture and Test Substances

The UM cell line 92.1 was kindly provided by Prof. Martine J. Jager (Leiden University Medical Center, Leiden, the Netherlands), in whose laboratory this cell line was established [68]. The 92.1 cells were authenticated by the profiling of short tandem repeats in previous studies [46,69] and harbor the Q209L-activating mutation in *GNAQ* [46,69]. The cells were grown under normoxic conditions at 37 °C with 5% CO_2_ in RPMI-1640 medium supplemented with 10% fetal bovine serum (FBS), 2 mM L-glutamine, 100 units/mL penicillin, and 100 µg/mL streptomycin (Life Technologies, Darmstadt, Germany) and passaged weekly via trypsinization.

Quercetin aglycone (Sigma-Aldrich, Darmstadt, Germany) was reconstituted in DMSO at 50 mM, stored as aliquots at −20 °C under light protection, and diluted in the culture medium at the indicated concentrations for the subsequent assays. The solvent controls were incubated with the culture medium that was supplemented with DMSO at the same volumes of quercetin. An additional group of cells were incubated in a low-serum medium with 0.5% FBS as a control for growth factor deprivation.

### 4.3. Immunocytochemistry

The cells were seeded into 8-well polycycloalkanes slides (Sarstedt, Nümbrecht, Germany) at a concentration of 10,000 cells/well, allowed to attach overnight, incubated with the test substances for 3 days at 37 °C with 5% CO_2_, fixed with 2% paraformaldehyde in phosphate-buffered saline (PBS) for 10 min followed by 4% paraformaldehyde–PBS for 10 min, rinsed three times in PBS, and kept in the blocking buffer (3% bovine serum albumin in 10 mM Tris-HCl (pH = 7.5), 120 mM KCl, 20 mM NaCl, 5 mM ethylenediaminetetraacetic acid, and 0.1% Triton X-100) for 30 min. The cells were incubated with the rabbit primary antibodies against Ki67 (Abcam, Cambridge, UK, 1:1000 dilution in blocking buffer), PFKP (Proteintech Germany GmbH, St. Planegg-Martinsried, Germany; 1:200 dilution in blocking buffer), or ZBTB20 (Proteintech Germany GmbH, 1:200 dilution in blocking buffer) overnight at 4 °C followed by the Cy3-conjugated secondary anti-rabbit antibodies (Jackson Immunoresearch, Cambridgeshire, UK, catalog number: 111-165-003, 3.75 µg/mL in blocking buffer) for 1 h at room temperature and protected from light. In the samples that were processed for the detection of PFKP or ZBTB20, actin filaments were stained with Alexa 488-phalloidin (Invitrogen, Thermo Fisher Scientific, Waltham, MA, USA; 240 units/mL in blocking buffer) for 30 min. The nuclei were counterstained with 0.5 µg/mL 4′,6-diamidino-2-phenylindole (DAPI, 0.5 µg/mL in PBS) for 10 min, with the rinsing of cells three times in PBS after each step. The cells were mounted in Mowiol and stored at 4 °C. Images were acquired with a monochrome digital camera (DFC 350 FX; Leica Microsystems, Wetzlar, Germany) that was attached to a fluorescence microscope (Leica DMI 6000B) by using Leica Application Software (Advanced Fluorescence 2.3.0, build 5131) and the following filter sets: A4 (excitation: 360/40 nm; emission: 470/40 nm); L5 (excitation: 460/40 nm; emission: 527/30 nm); and Y3 (excitation: 545/30 nm; emission: 610/75 nm).

### 4.4. MTT Assay and IC50 Calculation

The cells were seeded into 96-well plates (5000 cells per well), allowed to attach overnight at 37 °C, and incubated with the test substances in triplicate for three days at 37 °C. The MTT dye solution was added at a final concentration of 0.5 mg/mL into each well except for the background controls, and the cells were incubated further for 3 h at 37 °C. The formazan crystals were solubilized by DMSO, and the absorption at 544 nm was measured with a spectrophotometer (FLUOstar Optima, BMG Labtech GmbH, Ortenberg, Germany). The mean absorbance was calculated for each group after background substraction.

For the calculation of IC50, a scatter plot was constructed with the concentrations of the test substances versus the absorbance values on the x- versus y-axes, respectively, by using Windows Excel (Version 2402). The y-axis was converted into the logarithmic scale. The IC50 of quercetin was calculated from the function of the exponential trendline that was fitted onto the respective curve.

### 4.5. Live/Dead Assay

The cells were seeded into flat-based 96-well plates with black polystyrene frames (Lumox^®^ multiwell, Sarstedt, Germany) at a concentration of 8000 cells/well, allowed to attach overnight at 37 °C, and incubated with the test substances in quadruplicates for three days at 37 °C. The cells were then washed twice with PBS and incubated for 45 min at room temperature with the mixture of Calcein-AM and propidium iodide (ABP Biosciences, Beltville, MD, USA), which were diluted in PBS at the final concentrations of 2 µM and 4 µM, respectively. A single well per each group was left in PBS as the background control. The fluorescence intensities were measured by using a fluorometer (FLUOstar Optima) at the following wavelength settings: Calcein-AM—excitation: 485 nm, emission: 520 nm, and gain: 1100; propidium iodide—excitation: 544 nm, emission: 590 nm, and gain: 2300. The mean fluorescence in each group was calculated after substracting the corresponding background signals, and the ratio of the Calcein-AM/propidium iodide intensities was taken as the live/dead index per group. Prior to fluorescence microscopy, the nuclei were counterstained with 10 µg/mL DAPI in PBS for 10 min. Images of the cells were acquired by using an inverted fluorescence microscope (Leica DMI 6000B) by using the filter sets as described in the Immunocytochemistry section above.

### 4.6. BrdU Assay

The cells were seeded into 96-well plates at a concentration of 8000 cells/well, allowed to attach overnight, and incubated with the test substances for 3 days at 37 °C with 5% CO_2_. Incorporation of BrdU into the DNA was measured with a colorimetric detection kit (Abcam, Berlin, Germany) by following the manufacturer’s instructions. The cells were exposed to the BrdU reagent during the final 24 h of incubation. The cells that were incubated without the BrdU reagent served as the negative control. The absorbance values were read at 450 nm with a spectrophotometric microplate reader (FLUOstar Optima). The mean absorbance for each group was determined after subtracting the mean value of the negative control.

### 4.7. Measurement of Reactive Oxygen Species

The cells were seeded into flat-based 96-well plates with black polystyrene walls (Lumox^®^ multiwell, Sarstedt, Germany) at a concentration of 30,000 cells/well, allowed to attach overnight at 37 °C and treated with the test substances in triplicate for 12–13 h at 37 °C with 5% CO_2_. The cells were then incubated for 30 min at 37 °C with the MitoROS^TM^ 580 reagent that was diluted in the assay buffer following the manufacturer’s instructions (AAT Bioquest, Pleasanton, CA, USA). A single well per each group was left in normal test medium as the background control. The fluorescence intensities were measured from the well bottom by using a fluorometer (SpectraMax i3x; Molecular Devices, Munich, Germany) at an excitation of 540 nm and an emission of 590 nm, with a cutoff at 570 nm. For the normalization of cell numbers in each well, the cells were fixed with 4% paraformaldehyde in PBS for 10 min, washed twice briefly in PBS, counterstained with 0.5 µg/mL DAPI in PBS for 10 min, and rinsed twice in PBS. The fluorescence signals of the nuclear stainings were measured from the well bottom by using a fluorometer (SpectraMax i3x) at the excitation/emission values of 360/470 nm. For each group, the mean fluorescence values were calculated after substracting the corresponding background signals, and the intensity of the MitoROS signals was divided by the intensity of the respective nuclear staining for normalization based on the cell number. Images of the cells were acquired with the use of an inverted fluorescence miscroscope (Leica DMI 6000B) by using the A4 and Y3 filter sets as described in the Immunocytochemistry section above.

### 4.8. Glucose Uptake

The cells were seeded into 24-well plates at a concentration of 20,000 cells/well and allowed to attach overnight at 37 °C with 5% CO_2_. The cells were then rinsed twice with serum- and glucose-free RPMI-1640 medium and incubated for 30 min at 37 °C with the test substances in glucose-free RPMI-1640. Afterward, 6-NBDG was introduced into the culture medium at a final concentration of 300 µM, except for the negative control group, and the cells were further incubated for 30 min, followed by two brief rinses in PBS. The cells were then visualized immediately via fluorescence and phase-contrast microscopy (Leica DMI 6000B). Quantification of the integrated density was performed in a minimum of *n* = 201 cells/group using Fiji software (version 1.53t) [70]. For this purpose, the cell contours were circumscribed on the phase-contrast images. The mean intensity was determined by redirecting the measurements to the fluorescence images. The mean intensity of the negative control group was subtracted from the mean intensity of each cell, which was then multiplied with the cellular area for calculation of the integrated density.

### 4.9. Glycolysis Assay

The cells were seeded into flat-based 96-well plates with black, polystyrene walls (Lumox^®^ multiwell, Sarstedt, Germany) at a concentration of 60,000 cells/well and allowed to attach overnight at 37 °C with 5% CO_2_. The glycolytic rate was measured by using a fluorometric assay that was based on extracellular acidification following the manufacturer’s instructions (Abcam, catalog number: ab197244). Prior to the test, CO_2_ was removed by incubating the plate in a humidified, CO_2_-free incubator at 37 °C for 3 h to minimize background acidification. The cells were then treated with the test substances that were diluted in the respiration buffer of the assay in duplicate for 3 h at 37 °C without CO_2_ in a temperature-controlled fluorometer (SpectraMax i3x; Molecular Devices). The negative control of each group was incubated without the glycolysis assay reagent. The fluorescence intensity was measured at an excitation of 380 nm and an emission of 615 nm. The mean intensity of each group was calculated after background subtraction.

### 4.10. ATP Assay

The cells were seeded into 96-well plates at a density of 30,000 cells/well (100 µL/well), allowed to attach overnight, and incubated with the test substances in triplicate for 2 days. ATP levels were measured by using a luminescence assay as instructed by the manufacturer (Abcam, ab113849). As a control for equal cellular material, phase contrast images of the center of each well were acquired prior to cell lysis using an inverse microscope (Leica) at 50× magnification, and cell density was quantified by using Fiji software. The contents of each well were transferred into an opaque, white 96-well microplate, and luminescence was measured using a multi-mode microplate reader (SpectraMax i3x). After background subtraction, the mean ATP concentration of each group was normalized to the cell density of the corresponding group.

### 4.11. G6PDH Activity

The cells were seeded into 12-well plates at a concentration of 200,000 cells/well, allowed to attach overnight at 37 °C with 5% CO_2_, and incubated with the test substances for 11–13 h. G6PDH activity was measured by using a colorimetric assay in duplicate for each group following the manufacturer’s instructions (Cell Biolabs Inc., San Diego, CA, USA, catalog number: MET-5081). The background control of each group was treated without the G6PDH substrate. Absorbance at 450 nm was measured using a multi-mode microplate reader (SpectraMax i3x). After background subtraction, the mean G6PDH activity of each group was normalized to the total protein concentration of the corresponding cellular lysate that was determined by the bicinchoninic acid (BCA) assay (Thermo Fisher Scientific).

### 4.12. Total Glutathione Assay

The cells were seeded into 96-well plates at a density of 10,000 cells/well (100 µL/well), allowed to attach overnight, and incubated with the test substances in duplicate for 7–10 h. The wells without cells served as the background control. The total glutathione levels were measured by using a luminescence assay following the manufacturer’s instructions (Promega, Walldorf, Germany, V6611). The contents of each well were transferred into an opaque, white 96-well plate, and luminescence was measured by using a multi-mode microplate reader (SpectraMax i3x).

### 4.13. Immunoblotting

The cells that were grown in 6-well-plates were washed twice with ice-cold PBS, kept on ice, and lysed in ice-cold cell lysis buffer (50 mmol/L Tris-HCl, pH 7.4, 150 mmol/L NaCl, 2 mmol/L EDTA, 1% NP-40 [*v*/*v*], 0.1% SDS, 0.5% sodium deoxycholate, and 50 mmol/L NaF, supplemented with protease and phosphatase inhibitors (Sigma-Aldrich, Munich, Germany; 1:100 each) immediately before use) for 15 min under gentle agitation on a shaker. Lysates were cleared by centrifugation at 12,000× *g*, 4 °C for 20 min (Sigma 2-16PK; Hettich, Tuttlingen, Germany). The supernatants were collected and stored at −80 °C. Protein concentration was determined by the BCA assay (Thermo Fisher Scientific). Samples were denatured at 95–99 °C for 5 min and run in 4–10% TGX stain-free polyacrylamide gels (Bio-Rad, Munich, Germany) under denaturing and non-reducing conditions with 15 µg of protein/well. Protein loading in the gels was visualized using ChemiDoc MP stain-free imaging (Bio-Rad) through activation with ultraviolet light for 5 min. The gels were then equilibrated in blotting buffer (48 mmol/L Tris, 39 mmol/L glycine, 10% methanol [*v*/*v*], and 0.04% SDS [*w*/*v*]) for 10 min and transferred onto methanol-activated polyvinylidene fluoride (PVDF) membranes via semi-dry blotting (Biotec-Fischer, Reiskirchen, Germany) at a constant current of 0.8 mA/cm^2^ for 1 h. Protein transfer onto the membranes was visualized through the use of the ChemiDoc MP stain-free system. Membranes were blocked in 5% non-fat dry milk in Tris-buffered saline solution with 0.1% Tween-20 (TBST) for 1 h under gentle agitation and incubated with a mixture of the rabbit primary antibodies against ZBTB20 (Proteintech, 1:1000) and beta-actin (Abcam, 1:1000) that were diluted in TBST with 0.5% non-fat dry milk followed by the HRP-conjugated goat anti-rabbit secondary antibodies (Jackson Immunoresearch, Cambridgeshire, UK, 1:5000 in blocking buffer) for 1 h at room temperature. Signal detection was performed with the Super Enhanced Chemiluminescence Kit (ABP Biosciences, Rockville, MD, USA) using the ChemiDoc MP system. After stripping the membranes at 37 °C for 15 min as instructed by the manufacturer (Thermo Fisher Scientific), the blots were reprobed with the primary rabbit antibodies against PFKP (Proteintech, 1:2000), followed by the abovementioned secondary antibodies and signal detection.

### 4.14. Statistical Analysis

Data were analyzed using NCSS statistical software (Version 2021; NCSS, Kaysville, UT, USA) on Windows 10. The correlation between gene expression and the copy number of chromosome 3 was assessed through the use of the Pearson and Spearman tests. The normality of distribution and equality of variances were verified through the use of the Shapiro–Wilk and Brown–Forsythe tests, respectively. The association between binary factors and continuous variables was evaluated through the use of the non-paired two-sided *t*-test, assuming equal variance, or the Mann–Whitney U test with normal approximation. The categorical variables with three or more subgroups were analyzed using one-way analysis of variance or the Kruskal–Wallis test, with tie correction for the latter performed when necessary. Bonferroni correction was applied for multiple hypothesis testing. *p*-values less than 0.05 were considered as significant.

## 5. Conclusions

In conclusion, we report for the first time that aggressive UM cells with monosomy 3 exhibit a more glycolytic gene expression profile, which may confer these cells with a growth advantage by enabling the efficient usage of glucose for the production of energy or biomass depending on cellular priorities. Our preliminary findings also highlight the therapeutic potential of quercetin as a novel, affordable, and immediately available therapy approach to prevent the growth of UM cells by interfering with the glucose metabolism of these tumors. The anti-carcinogenic potential and safety of quercetin in UM therefore deserves further investigation in future studies by analyzing a wider range of quercetin concentrations for different cellular events and including multiple melanoma cell lines as well as normal melanocytes or skin cells as controls.

## Figures and Tables

**Figure 1 ijms-25-04292-f001:**
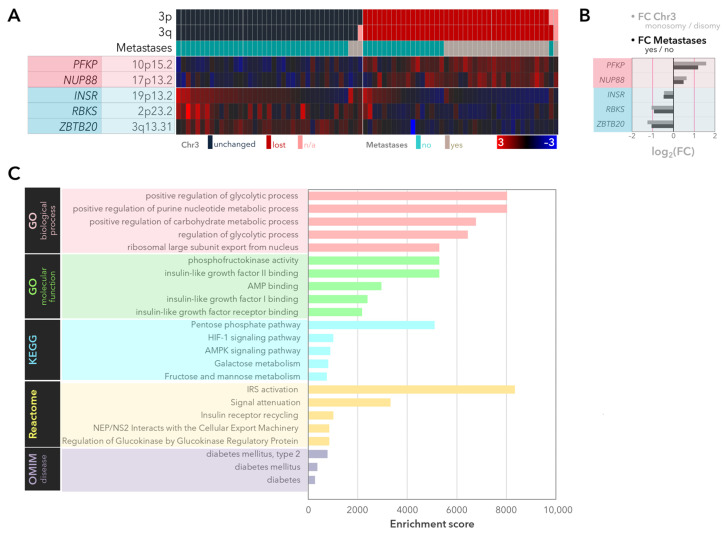
Differentially expressed genes in the monosomy 3 and metastatic tumors of the TCGA-UM cohort (*n* = 80 patients). (**A**) The tumor samples were aligned according to the copy numbers of chromosome 3p and 3q (dark blue: normal, red: loss) as well as the metastatic status in the uppermost three rows. The expression heatmap was constructed using the z-scores, with red and blue indicating mRNA levels that were up to three standard deviations above or below the mean (black), respectively. The gene symbols and loci are stated on the left. All of the genes had an adjusted *p*-value < 0.05 with regard to the copy number of chromosome 3 and metastases. n/a: Not available. (**B**) Fold changes (FC) of median gene expression in the monosomy 3 tumors and patients with metastases. The up- and downregulated genes are highlighted with a red or blue background, respectively. The red lines indicate an FC of |2|. Chr3: Chromosome 3. (**C**) Gene set enrichment analysis demonstrating the biological pathways and phenotypes that were over-represented among the differentially expressed genes. All of the *p*-values and false discovery rates were <0.05. For simplicity, the five most enriched processes or pathways were demonstrated for the Gene Ontology (GO), KEGG, and Reactome databases.

**Figure 2 ijms-25-04292-f002:**
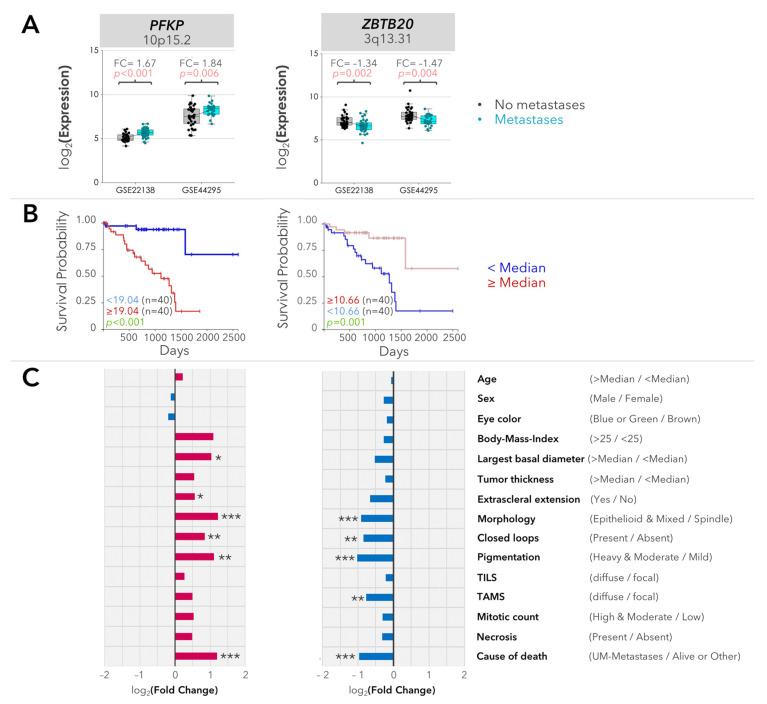
Validation of gene expression in two independent cohorts and the correlation of validated genes with prognostic factors. (**A**) The expression of *PFKP* and *ZBTB20* with regard to the metastases in the GSE22138 and GSE44295 cohorts is presented as boxplots. The mean values are connected by the sloped lines. For the GSE22138 cohort, the average expression of all of the available isoforms is presented (*n* = 2 isoforms for *PFKP*; *n* = 12 isoforms for *ZBTB20*). FC: fold change of median gene expression in the patients with metastases. The raw *p*-values were determined using the Mann–Whitney U test. Gene loci were indicated underneath the gene symbols. (**B**) The probability of overall survival with regard to the expression of *PFKP* and *ZBTB20* in the primary UMs of the TCGA cohort is demonstrated by the Kaplan–Meier curves. The median gene expression was taken as the cut-off value. *p*-values were determined via the log-rank test. (**C**) Expression of *PFKP* and *ZBTB20* with regard to the clinical and histopathological factors in the UM cohort of the TCGA study is presented as bar charts. The fold change of median gene expression in the subgroup with the unfavorable condition was calculated based on the classification of the prognostic factors according to the categories indicated in parentheses. The log_2_(Fold Change) values above or below 0 were depicted in pink or blue, respectively. TILS: tumor-infiltrating lymphocytes, TAMS: tumor-activated macrophages. The raw *p*-values were determined using the Mann–Whitney U test. * *p* < 0.05, ** *p* < 0.005, and *** *p* < 0.001.

**Figure 3 ijms-25-04292-f003:**
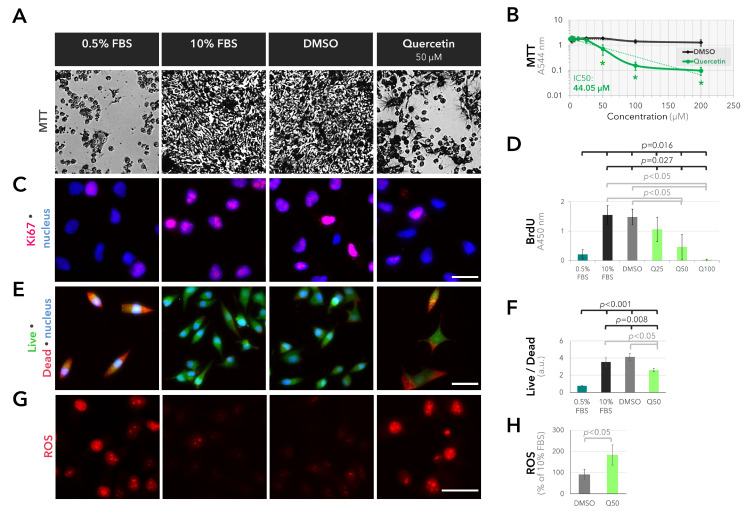
Quercetin suppresses the viability and proliferation of UM cells by increasing oxidative stress. (**A**) Representative light microscopy images of the UM cells after 3-day incubation with the substances indicated above followed by the MTT dye. A group of cells were incubated in culture medium with 0.5% fetal bovine serum (FBS) as the positive control for growth factor deprivation. A separate group of cells were incubated with the solvent of quercetin (dimethylsulfoxide; DMSO) at the same volume required for the administration of quercetin. (**B**) Quantification of the MTT assay (mean ± standard deviation (SD) of *n* = 4–5 independent experiments). The exponential trendlines were fitted and indicated in a dashed pattern in dark gray or green for DMSO or quercetin, respectively. Quercetin exhibited an IC50 of approximately 44.05 µM that was calculated from its respective trendline. * *p* < 0.05 for the pairwise comparison of the quercetin treatment to the corresponding solvent control, Mann–Whitney U test. (**C**) Representative images of immunofluorescence staining for the proliferation marker Ki67 (red) after 3 days. Nuclei were counterstained with DAPI (blue). Scale bar = 25 µm. (**D**) BrdU-assay demonstrating the dose-dependent anti-proliferative effect of quercetin after 3 days (mean ± SD of *n* = 3 independent experiments). *p*-values were determined through the use of the Mann–Whitney U or Kruskal–Wallis tests for pairwise or collective comparisons, respectively. (**E**) Representative images of the live/dead assay after 3 days. Scale bar = 50 µm. (**F**) Ratio of the live/dead intensity (mean ± SD of *n* = 3 wells from a representative experiment). *p*-values were assessed using the two-sided *t*-test or one-way analysis of variance for pairwise or collective comparisons, respectively. (**G**) Representative images demonstrating the accumulation of reactive oxygen species (ROS) in the cells treated with 50 µM quercetin for 12–13 h. Scale bar = 50 µm. (**H**) Quantification of the ROS intensity (mean ± SD of *n* = 3 independent experiments, Mann–Whitney U test).

**Figure 4 ijms-25-04292-f004:**
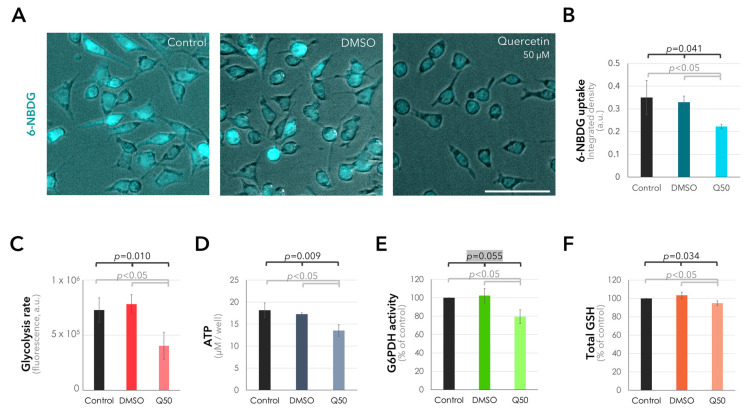
Quercetin interferes with glucose uptake and metabolism in the UM cells. (**A**) Representative images demonstrating the uptake of the fluorescent glucose analog 6-NBDG in the cells incubated with the normal medium alone (control) and with the supplementation of 50 µM quercetin or its solvent DMSO. The overlay images of fluorescence and phase-contrast microscopy are presented to demonstrate the cell boundaries. Scale bar = 100 µm. (**B**) Quantification of 6-NBDG uptake (mean ± SD of *n* = 3–4 independent experiments, a.u.: arbitrary units, Q50: 50 µM quercetin,). (**C**) Quantification of the fluorometric glycolysis assay, demonstrating the significant reduction in the glycolytic rate in response to quercetin after 3 h (mean ± SD of *n* = 3 independent experiments). (**D**) The luminescent ATP-assay, demonstrating the reduction in ATP levels in the cells incubated with 50 µM quercetin for 2 days (mean ± SD of *n* = 3 independent experiments). (**E**) Activity of the glucose-6-phosphate dehydrogenase (G6PDH), which functions as the rate-limiting enzyme of the PPP [17,20,21], as determined by a colorimetric assay after exposing the cells for 11–13 h to the indicated treatments (mean ± SD of *n* = 3 independent experiments). G6PDH activity was normalized to the amount of total protein extracted from each group. The insignificant *p*-value above 0.05 is highlighted with a gray background. (**F**) Quantification of the total glutathione (GSH) levels after 7–10 h (mean ± SD of *n* = 3 independent experiments). In panels (**B**–**E**), all of the pairwise comparisons were evaluated via the Mann–Whitney U test whereas the collective comparisons of the three subgroups were performed with the Kruskal–Wallis test (**B**,**E**,**F**) or one-way analysis of variance (**C**,**D**).

**Figure 5 ijms-25-04292-f005:**
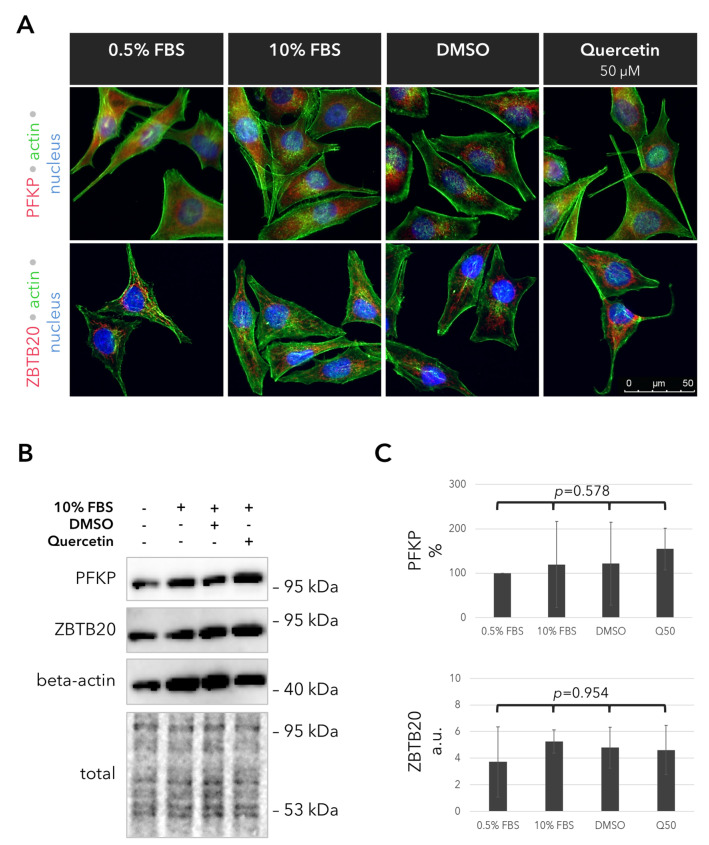
Expression of the PFKP and ZBTB20 proteins in response to quercetin after 1 day. (**A**) Representative images of the fluorescent immunostainings for PFKP and ZBTB20 (red). The actin filaments were visualized via Alexa 488-phalloidin staining (green). Nuclei were counterstained with DAPI (blue). (**B**) Immunoblotting for PFKP and ZBTB20. Quercetin was administered at a concentration of 50 µM. Membranes were probed for beta-actin as a loading control. The total amount of protein in each well is also depicted for a more comprehensive evaluation of sample loading. kDa: Kilodalton. (**C**) Quantification of the PFKP and ZBTB20 levels in the immunoblots, which were normalized to the total protein loadings in each well. The PFKP levels are stated as the percentage of the 0.5% FBS group. a.u.: arbitrary units. Data represent the mean ± standard deviation of *n* = 3 independent experiments. *p*-values were evaluated with the Kruskal–Wallis test for the simultaneous comparison of all subgroups.

## Data Availability

Publicly available datasets were analyzed in this study. The gene expression data of the UM cohort of the TCGA study were downloaded from the University of California, Santa Cruz Xena databank (https://xenabrowser.net/) (accessed on 23 April 2021). The validation cohorts can be accessed from the Gene Expression Omnibus databank (https://www.ncbi.nlm.nih.gov/geo/; accession numbers GSE22138 and GSE44295) (accessed on 10 May 2021).

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
