# Peer review of "Quercetin Impairs the Growth of Uveal Melanoma Cells by Interfering with Glucose Uptake and Metabolism"

_ijms, 2024, doi:10.3390/ijms25084292_

Round 1
Reviewer 1 Report (Previous Reviewer 2)
Comments and Suggestions for Authors
The study analyzed the influence of quercetin on glucose uptake and its metabolism in melanoma cells. In general, the topic of the study is significant and interesting. The authors have carefully prepared the manuscript. Both the introduction and the results were described accurately and clearly. The paper has been partially improved based on the reviewers' comments. However, certain issues could still be addressed (e.g., conducting analyses using a different melanoma cell line and normal skin cells, conducting studies using a greater range of quercetin concentrations). Nevertheless, it can be assumed that the research is preliminary and will be further continued. Therefore, I suggest that the authors discuss the limitations of the current study in the Discussion section and indicate directions for future experiments and analyses.
Author Response
Please see the attachment.

Reviewer 2 Report (Previous Reviewer 1)
Comments and Suggestions for Authors
Monosomy-3 in uveal melanoma (UM) is correlated with a higher risk of lethal metastases, e.g., to the liver. The development of UM can be potentially modulated by the glycolytic capacity. Initially, the authors analyzed the bioinformatic databases to asses the glycolytic profile of UM. The higher expression of phosphofructokinase P and the lower expression of ZBTB20 prompt metastasis and shorter overall survival of patients.
This research proposed quercetin as the compound with potential anti-cancer activity. Based on in vitro experiments in the UM cell line 92.1, i.e., cytotoxicity and ROS production were induced, glycolysis activity, pentose phosphate pathway, and ATP production were decreased.
The research is well-designed and interesting. It was shown that the glycolytic gene expression profile in UM cells enables their growth advantage by the possibility of the efficient usage of glucose for the production of energy or biomass. Quercetin can possibly be used in therapy to counteract the Warburg effect presented in most tumor cells. Moreover, active blood concentrations can be achieved by oral administration of quercetin, which creates the possibility of adjuvant treatment.
In my opinion, the limitation is the use of only one cell line. Moreover, the levels of PFKP and ZBTB20 proteins remained unchanged, therefore the manuscript does not provide a definitive explanation of the mechanism of action of quercetin. On the other hand, the authors discussed this and other limitations of the presented research very carefully. Therefore, the data presented may be sufficient as a preliminary result showing the first evidence that primary UMs with higher metastatic potential exhibit a more glycolytic gene expression profile.
Author Response
Please see the attachment.

This manuscript is a resubmission of an earlier submission. The following is a list of the peer review reports and author responses from that submission.
Round 1
Reviewer 1 Report
Comments and Suggestions for Authors
Monosomy-3 in uveal melanoma (UM) is correlated with a higher risk of lethal metastases, e.g., to the liver. The development of UM can be potentially modulated by the glycolytic capacity. Initially, the authors analyzed the bioinformatic databases to asses the glycolytic profile of UM. The higher expression of phosphofructokinase P and the lower expression of ZBTB20 prompt metastasis and shorter overall survival of patients.
This research proposed quercetin as the compound with potential anti-cancer activity. Based on in vitro experiments in the UM cell line 92.1, i.e., cytotoxicity and ROS production were induced, glycolysis activity, pentose phosphate pathway, and ATP production were decreased.
The research is well-designed and interesting. It was shown that the glycolytic gene expression profile in UM cells enables their growth advantage by the possibility of the efficient usage of glucose for the production of energy or biomass. Quercetin can possibly be used in therapy to counteract the Warburg effect presented in most tumor cells. Moreover, the active concentration in blood can be reached by quercetin oral administration.
The manuscript is well written. Limitations of the study were discussed by the authors. The methodology is varied and modern. Only some aspects need to be corrected/explained.
Major issues:
1. The statistical description of results is not always clear, e.g., when more than two bars are statistically compared, only one p-value is shown. In my opinion, each statistical comparison has an individual p-value. Please look at, Fig. 3D,E, 4B,C,D.
Fig. 4E,F - p-value not shown; what does an asterisk mean here?
Fig. 5C - one p-value was presented; what was compared?
2. Fig. 3H - p=0.049, why is the asterisk not used? The result is p<0.05
Minor issues:
1. Some editorial mistakes are present; for instance, in line 146 me-tastases, line 229 qlycolysis, line 244 "-" is unnecessary, and line 253 concen-tration.
2. Please check if gene names are always written in italics.
Reviewer 2 Report
Comments and Suggestions for Authors
The reviewed publication presents an analysis of the influence of quercetin on glucose uptake and its metabolism in melanoma cells. The authors focused on uveal melanoma as the research problem. The theoretical background of the research problem has been sufficiently presented and justified in the introduction and genetic cohort analysis. The selection of laboratory techniques is commendable and is one of the strengths of the work. The obtained results have been presented and described accurately and clearly. Overall, I consider the topic of the study to be significant and interesting. However, I must draw attention to the following issues:
1. In vitro experiments were unfortunately conducted only on one cancer cell line, which somewhat reduces the scientific value of the publication. Additionally, the authors did not choose to simultaneously conduct analyses using normal control cells (e.g., fibroblasts or melanocytes).
2. Most of the research was carried out using only one concentration of quercetin. In this situation, it is challenging to precisely determine the dependence of the observed effects on the dose or concentration of the investigated substance.
3. The justification for using such diverse cell incubation times with quercetin before measurements is not entirely clear. Some studies were conducted after 3 days of incubation, some after several hours, and a few analyses were performed after 30 minutes or a few hours of exposure. This leads to certain difficulties in the consolidation, analysis, and interpretation of the obtained results.
4. Please clarify how the calculation of the IC50 parameter was performed. From the graph, it seems that the value should be higher and hover around 100 µM.
Reviewer 3 Report
Comments and Suggestions for Authors
In this manuscript, the authors investigate the effect of quercetin on the growth of uveal melanoma (UM) cells. It was found that glycolysis and pentose phosphate pathway (PPP) genes are upregulated in UM cells with monosomy-3, known for their higher basal glucose uptake potential. Furthermore, the authors demonstrated that PFKP, a key glycolytic enzyme, is overexpressed, and the ZBTB20 gene is downregulated in patients with metastatic UM. The study also reveals that quercetin significantly impairs proliferation, glucose uptake, and glycolysis in the UM cell line 92.1. This research provides new insights into the effects of quercetin on UM cells. However, to enhance the study, the following points should be considered:
1. An investigation into the effect of quercetin on GLUT gene expression is necessary.
2. The use of additional UM cell lines is recommended for broader analysis.
3. The relationship between the quercetin results and other findings needs clearer elucidation in the results section.
Round 2
Reviewer 1 Report
Comments and Suggestions for Authors
Thank you for taking into account my suggestions.
I have no further comments.
Author Response
Thank you very much for your valuable comments and time to evaluate our manuscript. We are very pleased to have addressed your suggestions adequately.
Kind regards,
Aysegül Tura
Reviewer 3 Report
Comments and Suggestions for Authors
The authors have addressed the points which I noted.
Author Response

(The authors gave the same response as above.)
